## Pathogens

# Interferon-β induction heterogeneity during KSHV infection is correlated to levels and activation of the transcription factors ATF2 and RelA, and not IRF3

Machika Kaku[1,2], Marta Maria Gaglia [2*]

**1** Program in Immunology, Tufts Graduate School of Biomedical Sciences, Boston, Massachusetts, United States of America, **2** Department of Medical Microbiology and Immunology, Institute for Molecular Virology, and Carbone Cancer Center, University of Wisconsin - Madison, Wisconsin, United States of America

* marta.gaglia@wisc.edu

## Abstract

Careful regulation of type I interferons (IFN) like IFN-β is vital for balancing tissue damage and protection against infections. Heterogeneity in type I IFN expression among virally infected cells is a common phenomenon that may help limit IFN responses, but the source of this heterogeneity is poorly understood. We previously found that during Kaposi's sarcoma-associated herpesvirus replication, type I IFN induction was limited to a small percentage of infected cells. This heterogeneity was not explained by viral gene expression. Here, we used a fluorescent reporter and fluorescence activated cell sorting to investigate the source of the heterogeneity. Surprisingly, the canonical IFN induction pathway culminating in the activation of the IRF3 transcription factor was similarly activated between cells that made high vs. low/no IFN-β. In contrast, the activation or expression of the two other IFN transcription factors, the NF-κB subunit RelA and the AP-1 subunit ATF2, correlated with IFN-β induction. Our results suggest that during viral infection, activation of IRF3 does not automatically result in IFN responses at the level of individual cells, but that other factors, such as NF-κB and AP-1, are limiting for type I IFN induction.

## Author summary

The ability of mammalian cells to react to viral infections is a crucial step in the induction of immune responses. The first course of action for the cell is to express and release type I interferons like interferon-β (IFN-β), secreted molecules that warn surrounding cells. Single-cell level examination of gene expression has revealed that surprisingly, during many viral infections, only a small fraction of infected cells make IFN-β. This is likely a mechanism to prevent immune system overreactions. However, it remains unclear why only some cells respond. Here, we find that during

**Data availability statement:** Single-cell RNAseq used for analysis in Fig 5 was previously published and deposited in NCBI GEO under identifier GSE190558. All primary data are available on Figshare, DOI: doi.org/10.6084/m9.figshare.c.8132036.

**Funding:** This work was supported by National Institutes of Health (R01 CA268976 to M.M.G.) and by the University of Wisconsin-Madison, Office of the Vice Chancellor for Research and Graduate Education with funding from the Wisconsin Alumni Research Foundation (to M.M.G.). The funders had no role in study design, data collection and analysis, decision to publish, or preparation of the manuscript.

**Competing interests:** The authors have declared that no competing interests exist.

infection with Kaposi's sarcoma-associated herpesvirus, an oncogenic virus that affects immunocompromised individuals, the transcription factors AP-1 and NF-κB, rather than the more commonly studied IRF3, may decide which cells go on to make IFN-β. Our findings contribute to a better understanding of complex gene regulation and shed light on a process that fights an oncogenic virus.

## Introduction

Type I and III interferons (IFNs) such as interferon-β (IFN-β) and interferon-λs (IFN-λs) are potent cytokines that defend mammals against viral infections. As a key component of innate immunity, they are induced in response to pathogen-associated molecular patterns (PAMPs), such as nucleic acids with unfamiliar characteristics, or damage-associated molecular patterns (DAMPs) due to infection-induced stress. PAMPs and DAMPs are sensed by pattern recognition receptors (PRRs), leading to IFN induction. Type I and III IFNs then orchestrate antiviral defenses by signaling through the IFNAR1/2 and IFNLR1/IL10Rβ receptor complexes, respectively, and induce hundreds of interferon stimulated genes (ISGs) [1]. ISGs encode proteins that antagonize viral replication through a variety of mechanisms [2]. Despite this crucial antiviral role, IFNs are well known to be a double-edged sword because of their powerful effects on cell and tissue physiology. While insufficient IFN signaling leaves the host susceptible to infections, excessive IFN signaling causes tissue damage and contributes to autoimmune disease development [3–5]. Therefore, IFN-β and other type I and III IFNs are carefully regulated, and multiple host regulatory mechanisms have evolved to ensure their expression only under appropriate conditions.

An emerging dimension of IFN regulation is transcriptional heterogeneity at the single cell level. This aspect of IFN regulation has become clearer from the growing number of single cell RNA sequencing (scRNA-seq) analyses of infected cells. We and others have found only a small percentage (<10%) of virus-infected cells produce IFN-β or IFN-λs [6–21]. This has been reported in infections with the RNA viruses influenza A virus [7,10,11,14–17], Newcastle disease virus [18–20], West Nile virus [13], Sendai virus [21], and SARS coronavirus 2 [9], as well as the DNA viruses herpes simplex virus 1 [12] and human cytomegalovirus [8], and in our studies in Kaposi's sarcoma-associated herpesvirus (KSHV) [6]. We had previously discovered that KSHV hijacks the function of host caspases to prevent IFN-β induction when KSHV re-enters the replicative (lytic) cycle from a dormant (latent) infection state [22]. Adding a small molecule caspase inhibitor restores activation of the cytoplasmic DNA-sensing PRR cyclic GMP-AMP synthase (cGAS) [6] and results in IFN-β induction [22]. The amount of IFN-β produced under these conditions was substantial, according to ELISA measurements [22]. Nonetheless, subsequent scRNA-seq experiments revealed that the IFN-β comes from a small population of cells, with less than 4% of infected cells producing it even under this "derepressed" condition [6]. The limited number of type I/III IFN producing cells across a range of infections and cell types indicate that this is a feature of the IFN induction system.

Since subversion of type I IFN is a key evolutionary tactic for viruses in the virus-host arms race, heterogeneity in the expression and activity of IFN-inhibiting viral proteins has been proposed as the reason for heterogeneous IFN-β expression [15,20]. However, we and others have found that viral gene expression heterogeneity does not fully explain the observed IFN-β heterogeneity during infection with a range of different viruses – KSHV, influenza A virus, Sendai virus, and Newcastle disease virus [6,11,19–21]. In our scRNA-seq experiments with KSHV, IFN-β was only detected in cells in the early stages of the KSHV replication cycle after lytic reactivation, which made up 46% of the population, indeed suggesting a contribution from viral regulation of IFN [6]. Cells that are latently infected with KSHV likely do not detect the infection, while cells in later lytic stages have likely shut down host defenses. However, only 6% of the cells in the early lytic stages expressed IFN-β, demonstrating that the viral replication stage does not completely determine IFN induction [6]. Moreover, when we compared IFN-β positive and negative cells within this population, we did not find any viral gene that was specifically missing in IFN-β-producing cells, as would be expected for a heterogeneously expressed viral IFN suppressor [6]. These findings suggested that there is an additional, likely cellular, source of IFN-induction heterogeneity in this system, which also explains the consistency of this phenomenon across viral infections. The host may benefit from limiting the percentage of cells producing IFN-β in an infected population, as this may allow for optimal distribution of the cytokine and protection of surrounding cells without IFNs reaching dangerous levels. However, how IFN induction is restricted to a few cells remains unclear.

A potential mechanism could be restricted activation of one of the many signaling steps that are needed for type I and III IFN transcriptional activation downstream of PRRs. While different PRRs rely on different signaling adaptors, their signaling pathways generally converge on the kinase TANK-binding kinase 1 (TBK1), which phosphorylates the key transcription factor interferon regulatory factor 3 (IRF3). While IRF3 activation is commonly used as the main indicator for IFN-β transcription triggered by viral infection, IFN-β expression also requires additional transcription factor complexes: the nuclear factor kappa-light-chain-enhancer of activated B-cells (NF-κB) and the activator protein 1 (AP-1) complexes [23]. Together with IRF3, AP-1 and NF-κB make up the IFN-β enhanceosome. While the AP-1 and NF-κB complexes can be composed of many different members depending on the context, AP-1 complexes formed by c-Jun and activating transcription factor 2 (ATF2) and NF-κB complexes formed by the RelA/p65 and NFKB1/p50 subunits have been specifically linked to IFN-β transcription during viral infections [24]. Previous studies addressing IFN induction heterogeneity have proposed different explanations on how this pathway may result in heterogeneity. Some studies have suggested that promoter firing is inherently stochastic, i.e., that all the components of the induction pathway are active, including the transcription factors, but promoter firing only happens in some cells [19,25]. One study suggested this was due to differences in the 3D structure of the genome [25], while another was unable to pinpoint the source of the noise and concluded it may have to do with enhanceosome assembly [19]. In contrast, Zhao et al. proposed that every step leading to IFN induction is subject to heterogeneous noise, from viral replication to signaling protein activation to abundance of transcription factors [21]. However, Zhao et al. did not control for the fact that many genes encoding proteins in the IFN induction pathways are themselves ISGs, and this study also used murine cells which do not require NF-κB for IFN induction [21]. Another study used mathematical modeling to suggest IFN-β induction heterogeneity comes from the stochastic nature of signaling events such as the formation of macromolecule complexes and the distribution of transcription factors [20]. Overall, the cellular source of the IFN induction heterogeneity remains poorly defined.

In this study, we used the KSHV lytic infection and caspase inhibition system to further probe the source of heterogeneous IFN-β induction, as it is a robust and sustained system for inducing IFN in the context of a viral infection. We developed a reporter that allows us to separate cells expressing high vs. low levels of IFN-β and found that surprisingly, the canonical IFN induction pathway culminating in IRF3 activation was equally activated in both populations. However, activation or levels of other components of the IFN-β enhanceosome better correlated with IFN-β induction. In our system, activation of the NF-κB component RelA and baseline levels of the AP-1 component ATF2 may be the limiting factors for IFN-β transcription at the single cell level. Moreover, these findings suggest that that IRF3 serves as the infection signal, while

AP-1 and NF-κB provide the barrier to excessive IFN-β, at least in reactivating KSHV-infected BC-3 cells. This study highlights the nuanced mechanics behind the induction of a key component of our cells' antiviral response and identifies specific cellular factors that may contribute to limiting IFN-β induction to a small subpopulation of cells during KSHV infection.

## Results

### A fluorescent tdTomato reporter driven by a minimal human IFN-β promoter distinguishes cells that express high vs. low levels of IFN-β

Using scRNA-seq, we previously found that a very small fraction of KSHV-infected cells produce IFN-β when the virus is reactivated and caspases are inhibited [6]. In the scRNA-seq dataset, we calculated this fraction to be 3.8% of the total population and 6% of the infected cells in the early stage of the lytic cycle. Nonetheless, the rare IFN-β producing cells clustered together in our scRNA-seq data [6], suggesting that they have different characteristics from other cell populations. To identify factors involved in IFN-β regulation, we sought to separate and compare IFN-β producing and non-producing cells (Fig 1A and 1B). We previously engineered a tdTomato reporter driven by two copies of the 303 bp minimal promoter sequence from the human *IFNB1* gene (Fig 1A) and created stable BC-3 cell lines with this reporter (BC3-IFN-βp-tdTomato cells) [6]. We did not clonally select the cell line with the intention of focusing on IFN-β promoter sequence as the source of gene induction, and to eliminate contributions from the chromatin location of the reporter or variable reactivation of individual cell clones. BC-3 cells are latently KSHV-infected B cells isolated from a patient with primary effusion lymphoma [26]. The IFN-β promoter sequence was taken from previous luciferase or chloramphenicol acetyltransferase reporters [27–29]. tdTomato was selected because it is the brightest fluorescent protein whose signal is well separated from autofluorescence, which appears in the green channel and is high in BC-3 cells [30–33]. While our original scRNA-seq study was performed in KSHV-infected epithelial cells, the iSLK.219 system [6], we chose BC-3s for subsequent experiments because they are more biologically relevant, as B cells, not epithelial cells, are the target of long-term KSHV infection in patients. Moreover, the virus in iSLK.219 cells expresses GFP and RFP fluorescent reporters [34], which restricts our options for fluorescent markers. In BC-3 cells, the KSHV lytic cycle can be reactivated using the protein kinase C activator TPA. Treating BC3-IFN-βp-tdTomato reporter cells with TPA and the pan-caspase inhibitor IDN-6556 induced tdTomato expression compared to TPA treatment alone, with this expression limited to 2% of the cells when measured by flow cytometry [6]. Also, treating BC3-IFN-βp-tdTomato reporter cells with TPA and the pan-caspase inhibitor IDN-6556 induced IFN-β mRNA expression compared to no treatment, TPA alone, or the caspase inhibitor alone (Fig 1C).

To isolate cells expressing high and low IFN-β levels, we subjected the lytically reactivated BC3-IFN-βp-tdTomato reporter cells treated with caspase inhibitors to fluorescence-activated cell sorting (FACS) and collected tdTomato-positive and negative cells (Fig 1A and 1B). To confirm that tdTomato-based FACS results in the separation of cells based on IFN-β expression status, we measured tdTomato and IFN-β mRNA levels in the sorted samples. Importantly, we found that the endogenous IFN-β transcript was enriched in the tdTomato+ cells and depleted in the tdTomato- cells, as was tdTomato mRNA (Fig 1D and 1E). Because KSHV reactivation rates can be variable, we confirmed that all our experiments showed robust activation of the lytic cycle. We ensured that a high percentage of cells produced the KSHV lytic protein ORF45 (Fig 1F) and that there was clear induction of the early KSHV lytic gene ORF50 in cells treated with TPA or TPA and caspase inhibitors (Fig 1G). In addition, we measured a representative set of immediate early (S1A and S1C Fig), early (S1E and S1G Fig), and late (S1I and S1K Fig) KSHV genes in the IFN-β high and low sorted cells (i.e. tdTomato+ and tdTomato- cells). We saw no consistent difference in viral gene expression between the two populations, although the late genes showed at least a trend of de-enrichment in the IFN-β low cells (S1I and S1K Fig). This observation is consistent with our scRNA-seq data indicating that KSHV reactivation progression only broadly contributes to IFN-β induction, and that cells in the late stages of lytic reactivation do not produce IFN [6]. These results also demonstrate that KSHV reactivation progression does not fully explain IFN-β heterogeneity.

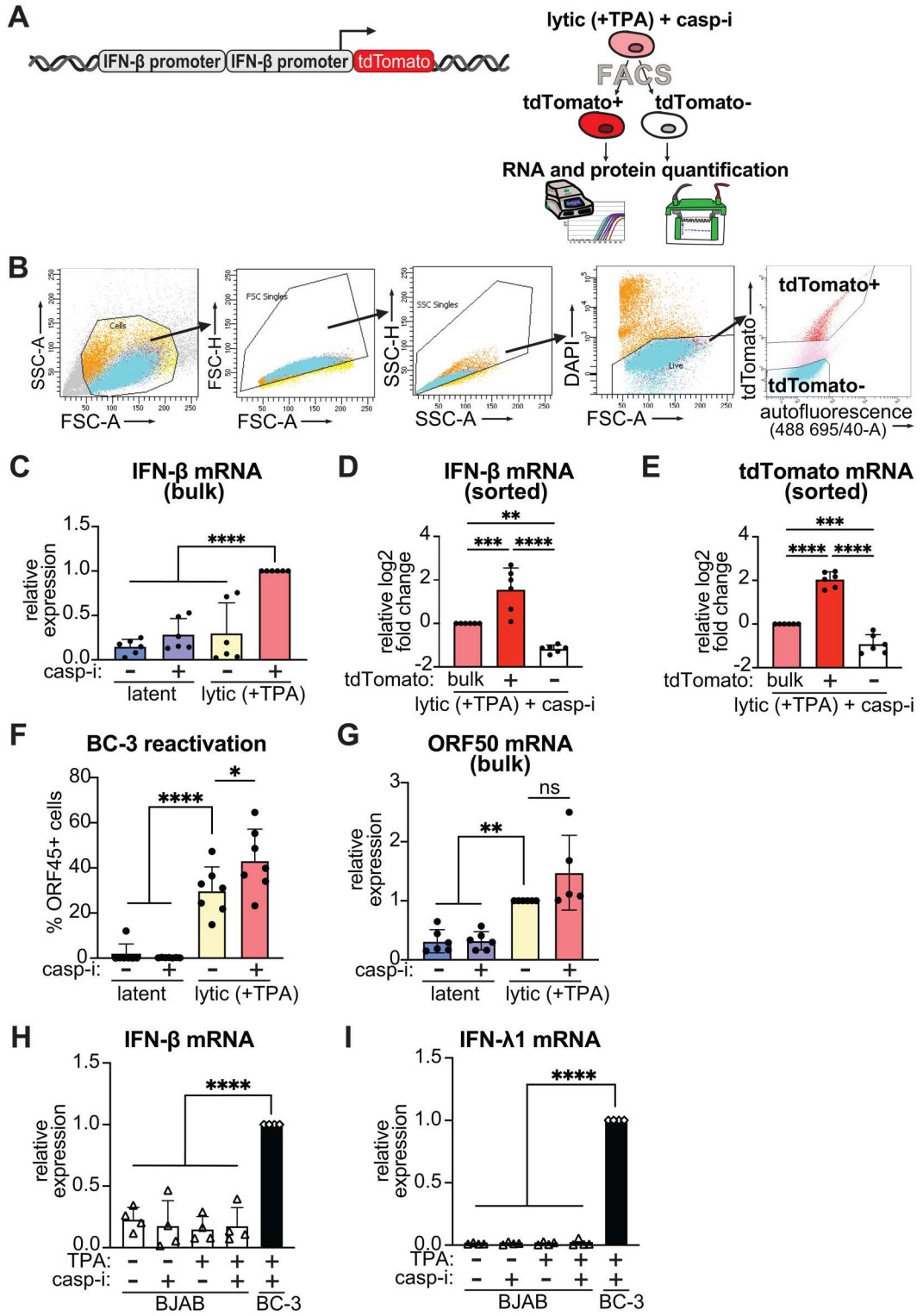

**Fig 1. An IFN-β-promoter driven fluorescent reporter allows isolation of cells expressing high and low levels of IFN-β. A.** Diagram of the IFN-βp-tdTomato reporter and sorting strategy. Two copies of the 303 bp minimal promoter sequence for the human IFNB1 gene were cloned in front of the tdTomato fluorophore [6]. This cassette was then introduced in BC-3 cells via lentiviral transduction for stable integration. Cells were sorted based on

tdTomato expression and RNA and protein samples were collected from the same experiment. **B-G**. BC3-IFN-βp-tdTomato reporter cells were treated with 20 ng/mL TPA for 48 hours to induce the lytic cycle ("lytic (+TPA)"). Where indicated, the cells were also treated with 10 µM of the pan-caspase inhibitor IDN-6556 ("casp-i"). For the lytic+casp-i sample, cells were collected without sorting ("bulk") or after sorting based on tdTomato expression (tdTomato+ and -). **B.** Gating strategy for FACS. **C-E, G.** IFN-β, tdTomato, and ORF50 mRNA levels were measured by RT-qPCR and normalized to 18S rRNA. For **C-E**, mRNA levels are plotted relative to the lytic+casp-i sample. For **G**, mRNA levels are plotted relative to the lytic sample. **F.** Cells were stained for KSHV ORF45 and analyzed by flow cytometry. **H-I.** KSHV-negative BJAB cells were treated with 20 ng/mL TPA for 48 hours. Where indicated, the cells were also treated with 10 µM of the pan-caspase inhibitor IDN-6556 ("casp-i"). BC-3 cells treated with 20 ng/mL TPA and 10 µM of the pan-caspase inhibitor IDN-6556 ("casp-i") were used as a positive control. IFN-β and IFN-λ1 mRNA levels were measured by RT-qPCR and normalized to 18S rRNA. mRNA levels are plotted relative to the BC-3 positive control. ns = $p > 0.05$, * = $p < 0.05$, ** = $p < 0.01$, *** = $p < 0.001$, **** = $p < 0.0001$, one-way ANOVA followed by Dunnett's (C, F, G, H, I) or Tukey's (D, E, G) post hoc multiple comparisons test. $n \geq 4$ for all experiments.

To ensure that our results were not confounded by lentiviral introduction of the reporter, we also generated and sorted BC-3 cells expressing tdTomato under the control of a CMV promoter (S2A Fig). As expected, in these cells the percentage of tdTomato-positive cells did not change with caspase inhibitor treatment (S2B Fig) and reactivation proceeded as expected in all conditions (S2C Fig). Importantly, the expression of the CMV promoter-driven tdTomato had no correlation to that of IFN-β (S2D and S2E Fig). The selective enrichment of IFN-β mRNA in the sorted cells when tdTomato was expressed from an IFN-β promoter but not a CMV promoter confirms that expression of the IFN-βp-tdTomato reporter is predictive of endogenous IFN-β induction. Moreover, it indicates that the heterogeneity is due to the IFN promoter activity, rather than variable expression due to random insertion of the transgene after lentiviral transduction.

While TPA-mediated reactivation is the most commonly used method of reliably reactivating KSHV in primary effusion lymphoma cell lines such as BC-3 cells [35,36], TPA has other pleiotropic effects such as activation of multiple signaling pathways involved in growth, apoptosis, and differentiation through activation of protein kinase C [37,38]. Therefore, to ensure that the use of TPA did not confound our studies, we measured IFN induction in an uninfected EBV and KSHV-negative B cell lymphoma line, BJAB cells [39]. BJAB cells are commonly used as an uninfected control for experiments with BC-3 cells and other primary effusion lymphoma cell lines [40,41]. We found that neither IFN-β nor the type III IFN IFN-λ1 were induced after TPA or TPA and caspase inhibitor (IDN-6556) treatment in uninfected BJAB cells (Fig 1H and 1I). Therefore, we concluded that the IFN induction in BC-3 cells was indeed due to KSHV lytic reactivation and virus sensing, not TPA addition, and that this system could be used for further experiments.

IFN-β signaling also potentiates its own transcription through upregulation of IFN-induction pathway components [42]. Indeed, in the scRNA-seq experiment, blocking IFN responses with an anti-IFN antibody cocktail reduced the number of IFN-β-positive cells [6]. Therefore, to specifically analyze the "first responder" cells in which IFN-β was only induced by pathogen sensing, without any feed-forward response, we treated the BC-3 cells with an anti-IFN antibody cocktail which blocks both type I IFNs and IFNAR. Consistent with the scRNA-seq results in iSLK.219 cells, we still observed IFN-β induction in the lytic BC-3 cells treated with caspase inhibitors after antibody treatment, although it was decreased (Fig 2A). FACS based on the tdTomato reporter also successfully enriched the rare IFN-β producing cells under these conditions (Fig 2B and 2C). We also still detected KSHV reactivation, measured by ORF45 staining or induction of the lytic gene ORF50 (Fig 2D and 2E), and a similar pattern of lytic gene expression in tdTomato+ and tdTomato- cells (S1B, S1D, S1F, S1H, S1J, and S1L Fig) in the presence of the antibody cocktail.

The enrichment of IFN-β mRNA in cells selected for tdTomato expression (with and without blocking IFN paracrine signaling) demonstrates that our IFN-βp-tdTomato reporter recapitulates heterogeneous expression of the native IFN-β transcript. High levels of reporter expression corresponded to high levels of IFN-β mRNA. Moreover, these results demonstrate that FACS of the BC3-IFN-βp-tdTomato reporter line allows us to separate cells producing high and low levels of IFN-β within a population and to analyze differences between them.

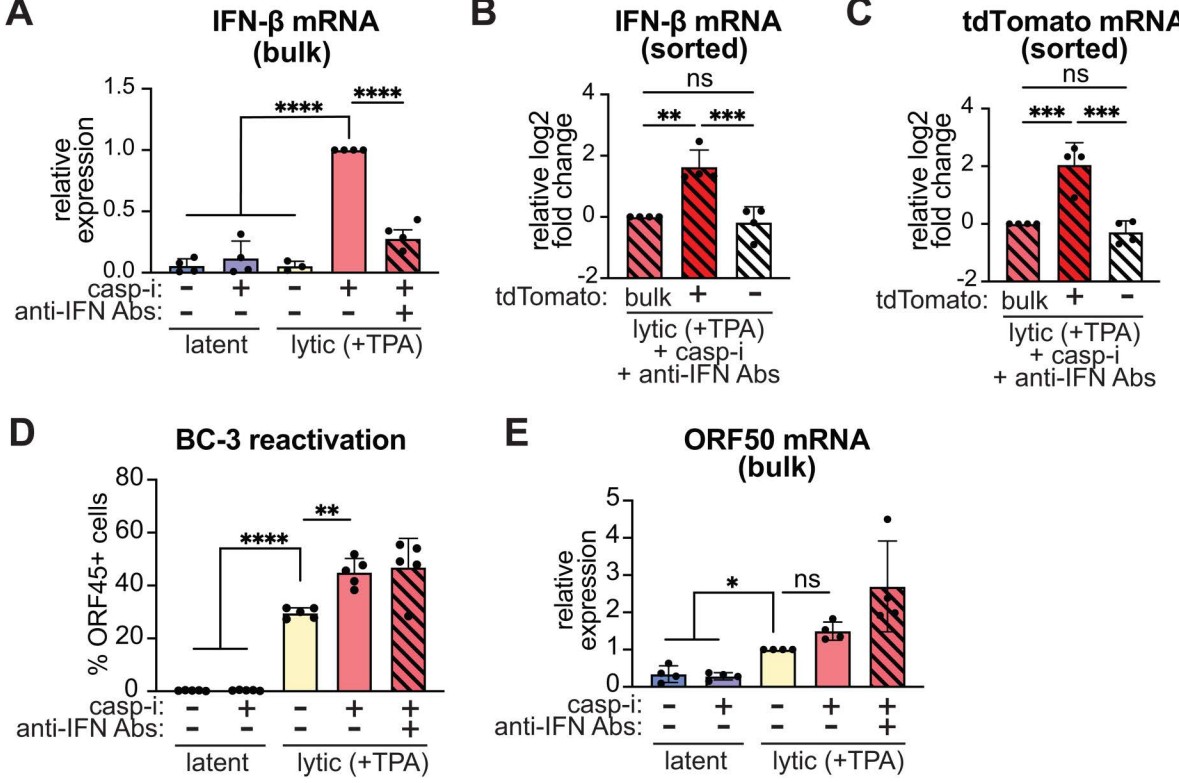

**Fig 2. The IFN-β reporter allows isolation of cells expressing high and low levels of IFN-β when paracrine signaling is blocked.** BC3-IFN-βp-tdTomato reporter cells were treated with 20 ng/mL TPA for 48 hours to induce the lytic cycle ("lytic (+TPA)"). Where indicated, the cells were also treated with 10 μM of the pan-caspase inhibitor IDN-6556 ("casp-i") and a cocktail of antibodies against type I IFNs and their receptor ("anti-IFN Abs") at 1:2000 dilution. For the lytic+casp-i+anti-IFN Abs sample, cells were collected without sorting ("bulk") or after sorting based on tdTomato expression (tdTomato+ and -). **A-C, E.** IFN-β, tdTomato, and ORF50 mRNA levels were measured by RT-qPCR and normalized to 18S rRNA. For **A**, mRNA levels are plotted relative to the lytic+casp-i sample. For **B** and **C**, mRNA levels are plotted relative to the lytic+casp-i+anti-IFN Abs bulk sample. For **E**, mRNA levels are plotted relative to the lytic only sample. **D.** Cells were stained for KSHV ORF45 and analyzed by flow cytometry. ns = p > 0.05, * = p < 0.05, ** = p < 0.01, *** = p < 0.001, **** = p < 0.0001, one-way ANOVA followed by Dunnett's (A, D, E) or Tukey's (B, C, F) post hoc multiple comparisons test. n ≥ 4 for all experiments.

## IFN-β is not stochastically induced

Some groups have proposed that heterogeneous expression of IFN-β may be generated by stochasticity of IFN-β promoter firing [19,25]. This model posits that the signals and activated proteins needed for IFN-β induction are present in all infected cells, but transcription of IFN-β only occurs stochastically in some cells. To test this hypothesis, we took advantage of the fact that pathogen sensing often induces type III IFNs like IFN-λ1 in addition to type I IFNs. The signaling cascade and transcription factors that regulate IFN-β and IFN-λ1 are largely the same [43], so any heterogeneity in upstream signaling should affect their induction in a coordinated way (Fig 3A). The IFN-β promoter is bound by AP-1, IRF3, and NF-κB, and the IFN-λ1 promoter by at least IRF3 and NF-κB [44]. However, since IFN-β and IFN-λ1 have separate promoters and are located on different chromosomes, stochastic noise in assembly of the transcription complexes or other aspects of promoter firing [19] would differentially affect them. Contrary to the stochasticity model, our results suggested that IFN-β and IFN-λ1 were induced in the same cells. In the previously published scRNA-seq results, IFN-λ1 was also induced in iSLK.219 cells upon lytic reactivation and caspase inhibitor treatment, and in the same rare cell populations that expressed IFN-β, indicating a shared regulation mechanism [6]. We have since performed further analysis of

PLOS Pathogens

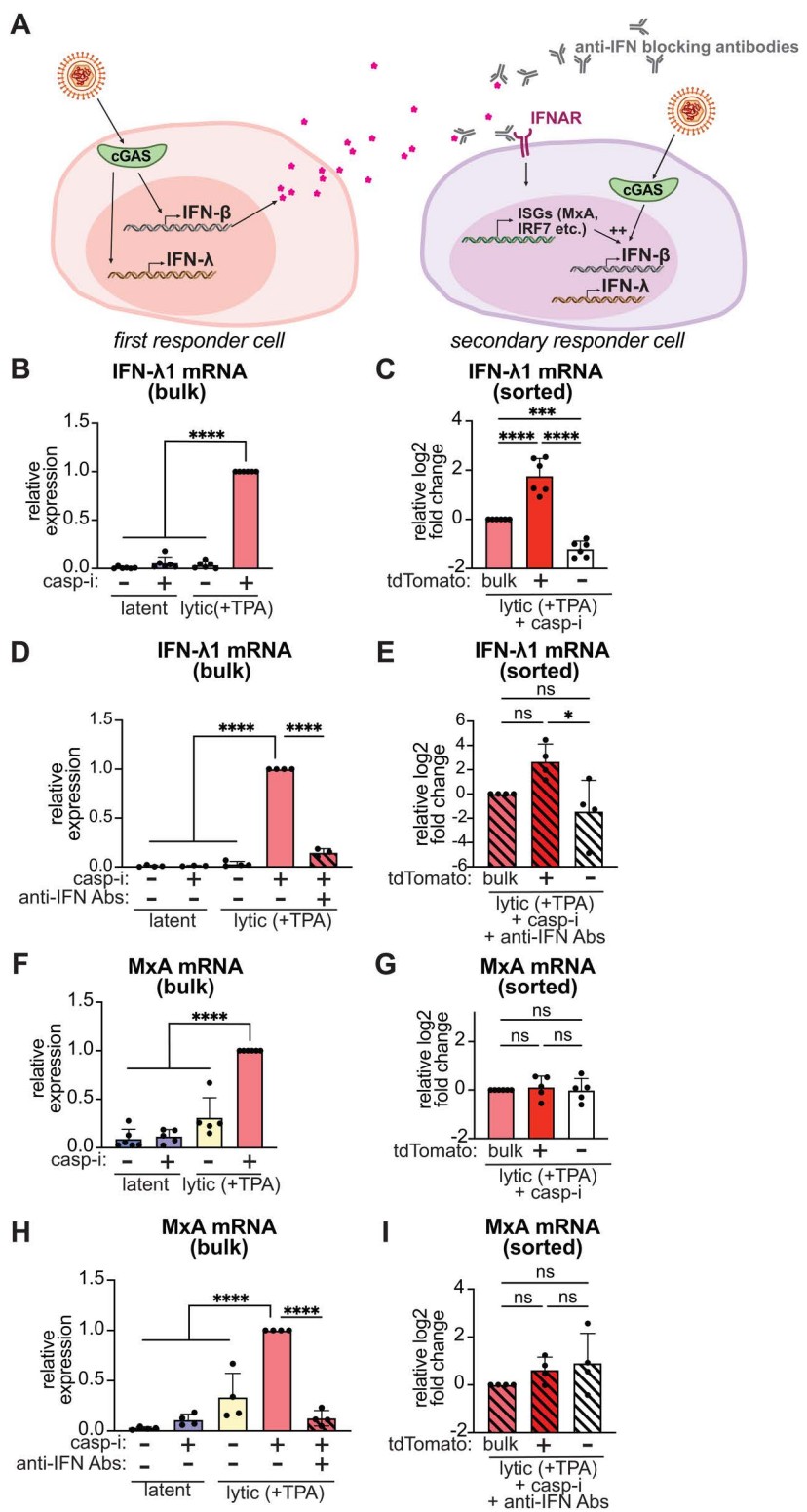

**Fig 3. IFN-λ1 and IFN-β are coordinately induced in a subset of cells, pointing to heterogeneity in the induction pathway rather than stochastic noise in expression. A.** Diagram of coordinate induction of IFN-β and IFN-λ. In "first responder" cells, IFN-β and IFN-λ are induced in response to infection or lytic reactivation downstream of PRRs. In "secondary responder" cells, ISGs such as MxA and many components of the IFN induction

pathway like IRF7 are induced in response to signaling downstream of IFN receptors. Induction of genes like IRF7 potentiates IFN induction in these cells. Anti-IFN antibodies block paracrine signaling from IFN-β, thus preventing ISG induction and potentiation of IFN induction in secondary responder cells. **B-I.** BC3-IFN-βp-tdTomato reporter cells were treated with 20 ng/ml TPA for 48 hours to induce the lytic cycle ("lytic (+TPA)"). Where indicated, the cells were also treated with 10 μM of the pan-caspase inhibitor IDN-6556 ("casp-i") and/or a cocktail of antibodies against type I IFNs and their receptor ("anti-IFN Abs") at 1:2000 dilution. For the lytic+casp-i (**C,G**) or lytic+casp-i+anti-IFN Abs (**E, I**) sample, cells were collected without sorting ("bulk") or after sorting based on tdTomato expression (tdTomato+ and -). IFN-λ1 (**B-E**) and MxA (**F-I**) mRNA levels were measured by RT-qPCR and normalized to 18S rRNA. In **B-D**, and **F-H**, IFN-λ1 and MxA mRNA levels are plotted relative to the lytic+casp-i sample. In **E** and **I**, IFN-λ1 and MxA mRNA levels are plotted relative to the lytic+casp-i+anti-IFN Abs bulk sample. Results in this figure were obtained from the same samples used in Figs 1 and 2 and are thus directly comparable to IFN-β enrichment shown in that Figure. ns=$p > 0.05$, *=$p < 0.05$, ***=$p < 0.001$, ****=$p < 0.0001$, one-way ANOVA followed by Dunnett's (B, D, F, H) or Tukey's (C, E, G, I) post hoc multiple comparisons test. n≥4 for all experiments.

the scRNA-seq data and found that indeed expression of IFN-λ1 and IFN-β were highly correlated at the single cell level. In fact, no other transcript showed closer correlation using the "FindAllMarkers" Seurat function for gene enrichment [45] on single cells separated by IFN-β expression. We also analyzed IFN-λ1 expression after sorting BC3-IFN-βp-tdTomato cells, using the same samples in which we analyzed IFN-β expression in Fig 1. Like IFN-β, IFN-λ1 was induced in BC-3 lytic cells treated with the caspase inhibitor IDN-6556 (Fig 3B). Moreover, IFN-λ1 was enriched and depleted in concert with IFN-β in the tdTomato+ and tdTomato- cells, respectively (Fig 3C. compared to IFN-β in the same samples, Fig 1D). This result indicates that IFN-β and IFN-λ1 were expressed in the same subset of cells. Similarly to IFN-β, IFN-λ1 induction can also be potentiated in response to IFN signaling by a feed-forward loop (Fig 3A), so we repeated the experiments in the presence of anti-IFN antibodies to block paracrine signaling and checked that the correlation was not a response to IFN. While this treatment did reduce IFN-λ1 induction dramatically, the pattern of IFN-λ1 and IFN-β co-expression was also observed in the absence of IFN responses, as both the IFN-β (Fig 2B) and IFN-λ1 (Fig 3E, same samples) transcripts were enriched in the tdTomato+ cells. These results stand in contrast to MxA, a classical ISG. MxA was induced in the lytically reactivated and caspase inhibitor-treated cells (Fig 3F), i.e. under conditions where IFNs are secreted, as expected for an ISG. Moreover, also as expected, anti-IFN antibodies blocked MxA induction (Fig 3H; anti-IFN antibodies were only added to the lytic+caspase inhibitor condition, as this is the condition in which we see high IFN production). This result also confirms that the antibody treatment blocks IFN responses. However, unlike IFN-λ1, MxA was not enriched in sorted cells in the presence or absence of paracrine signaling, demonstrating that the coordinate expression of IFN-λ1 and IFN-β was not related to IFN responses and IFN signaling potentiation of IFN-λ1 induction (Fig 3G and 3I). These results are consistent with the scRNA-seq data in iSLK.219 cells, where we detected IFN-induced ISG expression in all cell clusters, not just the ones producing IFN-β, whereas IFN-λ1 was mostly detected in IFN-β-producing clusters [6]. Therefore, we conclude that IFN-β and IFN-λ1 are not stochastically transcribed but coordinately regulated. Moreover, the fact that they are expressed in the same rare subset of KSHV-infected cells suggests that heterogeneity in the same upstream determinants leads to cell-to-cell differences in type I and III IFN induction.

## TBK1 and IRF3 activity does not explain IFN-β heterogeneity

The results above suggested that activation of the IFN-β induction pathway, rather than IFN-β transcription, was heterogeneous. However, there are several signaling steps in the IFN-β induction pathway that could be the bottleneck determining whether cells transcribe IFN-β. Therefore, we monitored activation of the IFN-β induction pathway. Using inhibitors for cGAS and TBK1, we confirmed that IFN-β and IFN-λ1 induction was driven by the cGAS-STING pathway through the TBK1 kinase in KSHV-reactivated BC-3 cells (Fig 4A and 4B), similarly to what we had previously found in the epithelial iSLK.219 cells [6,22]. In addition, in this system the cGAS ligand is likely mitochondrial DNA, because depletion of mitochondrial DNA with 2'3'-dideoxycytosine significantly reduced IFN-β induction despite no change in KSHV reactivation (S3 Fig). We thus tested whether TBK1 was activated only in the cells that induced IFN-β by comparing levels of TBK1 phosphorylation in lytically reactivated, caspase inhibitor-treated cells sorted

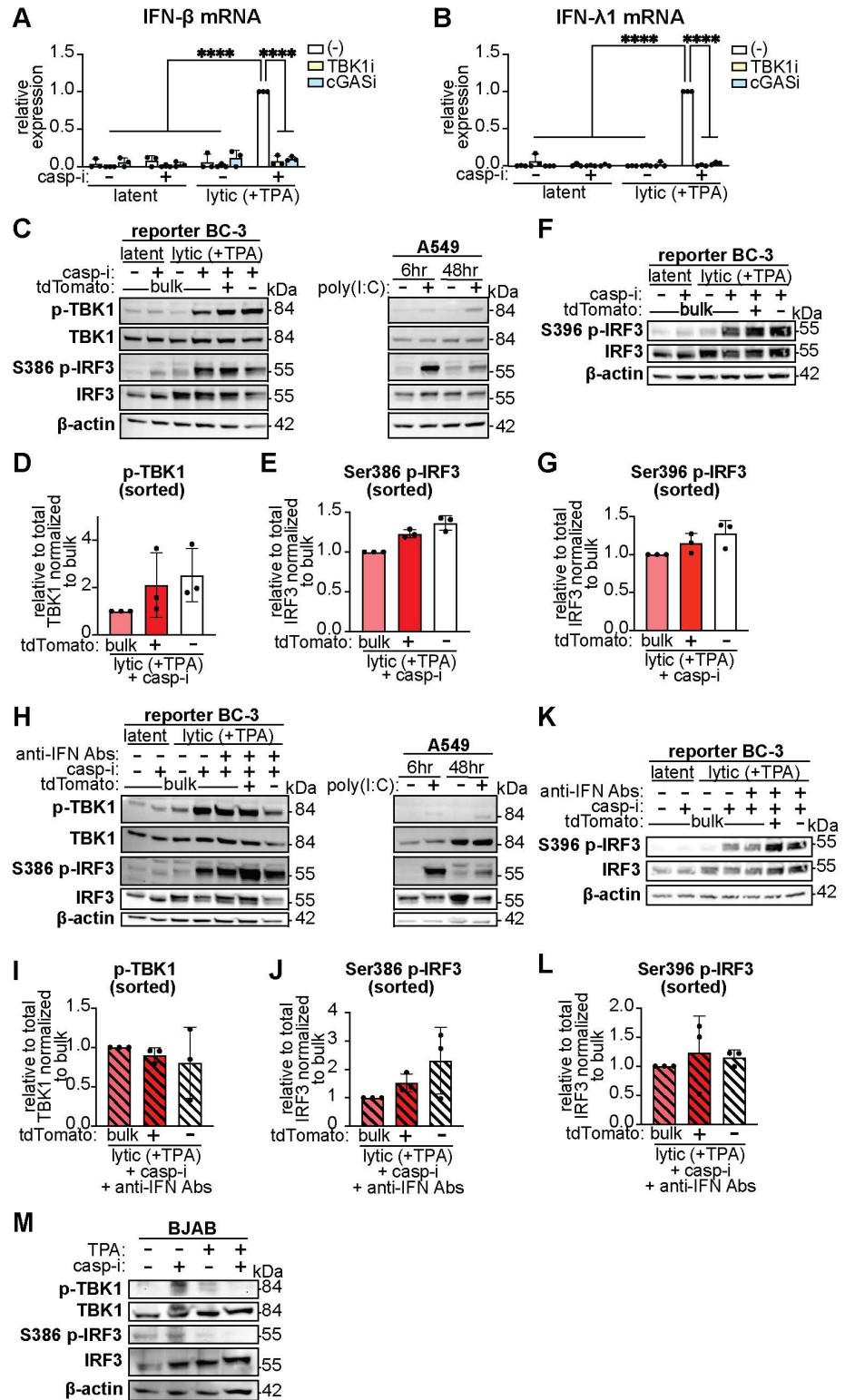

**Fig 4. TBK1 and IRF3 are necessary but not sufficient to direct IFN-β expression, as they are activated even in cells with low IFN-β expression. A-L.** BC3-IFN-βp-tdTomato reporter cells were treated with 20 ng/ml TPA for 48 hours to induce the lytic cycle ("lytic (+TPA)"). Where indicated, the cells were also treated with 10 μM of the pan-caspase inhibitor IDN-6556 ("casp-i"), 10 μM of the TBK1 inhibitor MRT76307 ("TBK1i"), 10 μM of the

cGAS inhibitor G140 ("cGASi"), and/or a cocktail of antibodies against type I IFNs and their receptor ("anti-IFN Abs") at 1:2000 dilution. **A-B.** IFN-β and IFN-λ1 mRNA levels were measured by qRT-PCR and normalized to 18S rRNA. The expression levels are plotted relative to lytic+casp-i samples that did not receive TBK1i or cGASi treatment for each experiment. n = 3. **** = p < 0.0001. Two-way ANOVA followed by Tukey's post hoc multiple comparisons test. **C-L.** Protein lysates were probed for phosphorylated IRF3 (Ser386 or Ser396), total IRF3, phosphorylated TBK1 (Ser172), total TBK1, and β-actin as a loading control. As a positive control for IRF3 and TBK1 activation/phosphorylation, A549 cells were treated with 7 μg/mL poly(I:C) for 6 and 48 hours before protein lysates were collected. Protein was isolated from treated BC3-IFN-βp-tdTomato reporter cells without sorting ("bulk"), or after sorting the lytic+casp-i (**C-G**) or lytic+casp-i+anti-IFN Abs (**H-L**) sample based on tdTomato expression. Western blot for phosphorylated TBK1 and phosphorylated IRF3 for bulk lytic+casp-i or lytic+casp-i+anti-IFN Abs treated samples and sorted tdTomato+ and tdTomato- samples were quantified by normalizing the density of phosphorylated protein to unphosphorylated protein bands. Phosphorylation levels are plotted relative to the bulk lytic+casp-i or lytic+casp-i+anti-IFN Abs treated samples. **M.** KSHV-negative BJAB cells were treated with vehicle, 20 ng/ml TPA, and/or 10 μM of the pan-caspase inhibitor IDN-6556 ("casp-i") for 48 hours. Protein was isolated and stained for phosphorylated TBK1, total TBK1, phosphorylated IRF3 (Ser386), and total IRF3. Images shown are representative of 3 replicates. None of the differences in quantified levels are statistically significant based on one-way ANOVA followed by Tukey's post hoc multiple comparisons test (D-E, G, I-J, L).

based on tdTomato status. To ensure the antibodies detected the correct epitopes, we used poly(I:C)-treated A549 cells as positive controls for TBK1 phosphorylation. In every experiment, tdTomato-based sorting resulted in the expected enrichment and depletion of IFN-β mRNA in tdTomato+ and tdTomato- cells, respectively (RNA measurements are shown in Fig 1D). However, surprisingly, we observed no difference in activated (phosphorylated) TBK1 between IFN-β high and low cells (Fig 4C and 4D). Since TBK1 phosphorylates and activates the well-characterized IFN-β transcription factor IRF3, we also tested whether IRF3 was differentially activated in the IFN-β high vs. low cells (i.e. tdTomato+ and tdTomato-), again using poly(I:C)-treated A549 cells as a positive staining control. Similarly to TBK1, we observed no difference in IRF3 phosphorylation at serine 386 and serine 396, two key sites that play synergistic roles in IRF3 activation [46] (Fig 4C and 4E-G). We also observed no difference in TBK1 and IRF3 phosphorylation between IFN-β high and low cells when we blocked autocrine and paracrine IFN-β signaling with anti-IFN antibodies (Fig 4H-L). These analyses were also conducted on samples in which we had verified IFN-β mRNA enrichment in the tdTomato+ sorted cells (Fig 2B). Therefore, while both TBK1 and IRF3 were active in lytic cells treated with caspase inhibitors with or without anti-IFN antibodies, their activation alone was not predictive of IFN-β induction. To ensure that the lytic cycle-inducing TPA treatment was not a confound for these experiments, we treated uninfected BJAB cells with the same drugs (TPA and the caspase inhibitor IDN-6556). We confirmed that these treatments did not activate TBK1 and IRF3 in the absence of KSHV infection (Fig 4M). Collectively, the results in Fig 4 indicate that the canonical IFN induction pathway is activated in many, likely most, lytically reactivated KSHV-infected BC-3 cells when caspases are inhibited. However, an additional factor besides IRF3 activation keeps IFN-β induction limited to a small fraction of the population.

### Expression of the AP-1 subunit ATF2 and activation of the NF-κB subunit RelA correlate with IFN-β induction and may drive heterogeneity

IRF3 activation is assumed to lead to IFN-β transcription, and phosphorylation of IRF3 is commonly used as a proxy for IFN induction. However, the IFN-β enhanceosome contains two additional transcription factor complexes: AP-1, composed of c-Jun and ATF2, and NF-κB, composed of RelA/p65 and NFKB1/p50 (Fig 5A) [24,47,48]. NF-κB and IRF3 are also verified transcription factors for IFN-λ1 [49]. How AP-1 and NF-κB are activated during IFN-β induction and their relation to activation of cGAS and other PRRs is not well understood. Given that IRF3 activation did not explain IFN-β heterogeneity, we considered the possibility that AP-1 and/or NF-κB might be determinants of heterogeneous IFN-β induction in our system. To verify the importance of AP-1 and NF-κB for IFN-β induction in our system, we inhibited them with SP100030, a dual AP-1 and NF-κB inhibitor. We found that SP100030 significantly reduced IFN-β induction in our system without impacting lytic reactivation, demonstrating that these factors play a key regulatory role (S4A and S4B Fig). To further ensure that our reporter was responsive to all three components of the IFN-β enhanceosome, AP-1, IRF3, and NF-κB,

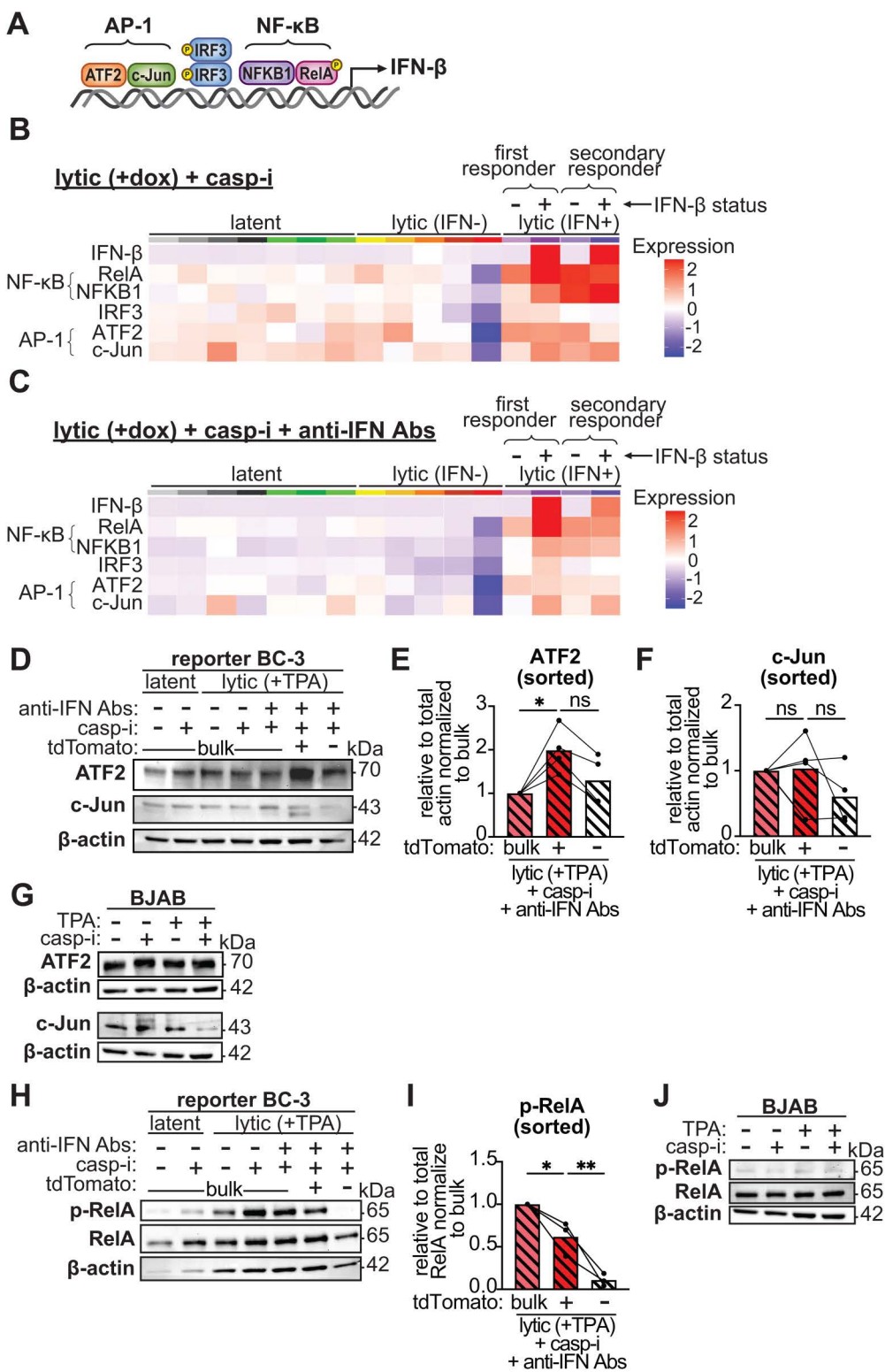

**Fig 5. Levels of the AP-1 subunit ATF2 and phosphorylation of the NF-κB subunit RelA are higher in cells with high IFN-β expression. A.** Diagram of the IFN-β enhanceosome factors and their position on the IFN-β promoter. In the IFN-β enhanceosome, NF-κB is composed of NFKB1 (p50) and RelA (p65), and AP-1 is composed of ATF2 and c-Jun. **B-C.** Re-analysis of single-cell RNA sequencing (scRNA-seq) data in Tabtieng, Lent, et al

2022 [6]. For this dataset, iSLK.219 cells were treated with 1 µg/mL doxycycline for 4 days to induce the lytic cycle ("lytic"). The cells were also treated with 10 µM of the pan-caspase inhibitor IDN-6556 ("casp-i"), as well as a mixture of antibodies against type I IFNs and its receptor ("anti-IFN Abs") (**C** only) to block IFN-β paracrine signaling. The average mRNA levels of genes in the IFN-β enhanceosome from the scRNA-seq data are shown for each of the clusters of cells identified by the original analysis (indicated by different colors above the heatmap). Clusters were classified as "latent", "lytic", or "lytic (IFN+)" based on the pattern of viral genes they expressed and whether they expressed IFN-β. For "lytic (IFN+)" clusters, cells were divided in "first responder" or "secondary responder" based on the change in IFN-β levels once the anti-IFN antibodies were added. The "lytic (IFN+)" clusters were also further separated into two groups based on the IFN-β status of individual cells. **D-F, H-I.** BC3-IFN-βp-tdTomato reporter cells were treated with 20 ng/ml TPA for 48 hours to induce the lytic cycle ("lytic (+TPA)"), 10 µM of the pan-caspase inhibitor IDN-6556 ("casp-i"), and where indicated, a cocktail mix of antibodies against type I IFNs and their receptor ("anti-IFN Abs") to block IFN-β paracrine signaling. Protein was isolated from BC3-IFN-βp-tdTomato reporter cells without sorting ("bulk"), or after sorting the lytic+casp-i+anti-IFN Abs based on tdTomato expression. Western blots were probed for ATF2 (**D**), c-Jun (**D**), total and phosphorylated RelA (Ser536) (**H**), and β-actin as a loading control. Levels of ATF2 (**E**) and c-Jun (**F**) in sorted samples were quantified from the western blots, normalized to β-actin, and plotted relative to bulk lytic+casp-i+ anti-IFN Abs samples. Levels of phosphorylated RelA (Ser536) (**I**) were quantified by normalizing to total RelA and plotted relative to bulk lytic+casp-i+ anti-IFN Abs samples. **G, J.** KSHV-negative BJAB cells were treated with vehicle, 20 ng/ml TPA and/or 10 µM of the pan-caspase inhibitor IDN-6556 ("casp-i") for 48 hrs. Protein was isolated and stained for ATF2 (**G**), c-Jun (**G**), phosphorylated (Ser536) and total RelA (**J**), and β-actin as a loading control. Images are representative of 4 replicates (**D**), 3 replicates (**H**), or 2 replicates (**G, M**). ns=$p > 0.05$, *=$p < 0.05$, **=$p < 0.01$, one-way ANOVA followed by Tukey's post hoc multiple comparisons test.

we mutated their respective binding sites and verified that the tdTomato mRNA was only induced from the non-mutated promoter in lytic+caspase inhibitor-treated BC-3 cells, despite similar levels of reactivation (S4C-E Fig).

We previously reported that expression of NF-κB family members was enriched in IFN-β-expressing cells based on scRNA-seq analysis of KSHV-infected epithelial cells treated with caspase inhibitors [6]. We reanalyzed the scRNA-seq dataset to test whether this was also the case for the AP-1 components. To further explore the relationship between IFN-β and these transcription factors, we separated the cells from the two IFN-β expressing clusters by IFN-β status. The AP-1 component ATF2 was enriched in both IFN-β-positive and negative cells within the IFN-β expressing clusters, while the AP-1 component c-Jun and the NF-κB components RelA and NFKB1 were specifically enriched in the IFN-β-positive cells (Fig 5B). These enrichments were particularly evident when anti-IFN antibodies were added to block paracrine and autocrine signaling, because most of these genes were also induced by IFN-β signaling (Fig 5C). To note, there was an enrichment of IRF3 mRNA in the IFN-β-producing cells in the scRNA-seq data (Fig 5B and 5C), even though we did not detect enrichment at the protein level in BC-3 cells (Fig 4C, 4F, 4H, and 4K). We also observed that after blocking paracrine signaling, IFN-β expression was reduced in one of the IFN-β expressing clusters (Fig 5B and 5C). This cluster may include mostly "secondary responder" cells in which IFN-β is induced mainly after the initial IFN signaling potentiates the sensitivity of the cells (Fig 3A). In contrast, the other cluster likely comprises "first responder" IFN-β-producing cells (Fig 3A). These results are consistent with the overall reduction in IFN-β induction that we saw upon addition of the anti-IFN antibodies (Fig 2A).

We then tested whether NF-κB and AP-1 were similarly enriched at the protein level in BC-3 cells expressing high levels of IFN-β. To reduce confounds from paracrine signaling observed in the scRNA-seq results, we performed the FACS experiments in the presence of anti-IFN blocking antibodies. Interestingly, we observed an enrichment in the AP-1 component ATF2 in tdTomato+ cells, indicating that ATF2 levels were higher in cells that expressed high levels of IFN-β (Fig 5D and 5E). Total ATF2 levels did not change between conditions in bulk samples, indicating that ATF2 expression was not induced at the bulk level during infection and/or by caspase inhibitor treatment. Instead, these results suggest that basal expression of ATF2 may be variable and that IFN-producing cells had higher ATF2 levels prior to lytic reactivation (Fig 5D). It is also possible that ATF2 levels are upregulated only in the rare cells that express IFN. As expected, ATF2 levels were not changed by TPA and/or caspase inhibitor treatment in uninfected BJAB cells (Fig 5G). Levels of the other AP-1 component, c-Jun, were variable across replicates, but in two out of three replicates a unique shorter isoform was detected in the sorted IFN-β high cells (Fig 5D). We also stained for the phosphorylated forms of ATF2 and c-Jun. We did not observe phosphorylation of these proteins at the bulk level after lytic reactivation and caspase inhibitor treatment, nor did we observe a difference in the sorted samples (S5A-C Fig).

IFN-β high cells were also enriched in phosphorylated RelA, one of the NF-κB components, whereas there was little to no phosphorylated RelA in the IFN-β low cells (Fig 5H-I). This result points to RelA activation only in IFN-β high cells. RelA phosphorylation was not induced by TPA and/or caspase inhibitor treatment in uninfected BJAB cells at the same time-point (two days after TPA addition), indicating this was specific to infected cells (Fig 5J).

To further examine AP-1 and NF-κB activity in the sorted cells, we quantified expression of select AP-1 and NF-κB targets. We detected enrichment of some of the tested targets in IFN-β high, tdTomato-positive cells: ATF3, an AP-1 target, and TNFAIP3, a target of both AP-1 and NF-κB (S5D-E Fig). This enrichment supports the idea that AP-1 and NF-κB are more active in the rare cells that induce IFN-β. Overall, these results indicate that ATF2 and p-RelA likely have a dominant role in cell-specific IFN-β induction. Particularly elevated ATF2 levels may be needed for IFN-β induction, while RelA is likely activated in response to a stimulus to potentiate IFN transcription in the small subset of IFN producing cells (Fig 6).

## Discussion

In this study, we investigated underlying differences between cells that induce high vs. low levels of IFN-β in the context of KSHV infection and caspase inhibition. We reveal that the IFN-β enhanceosome components AP-1 and NF-κB, but not

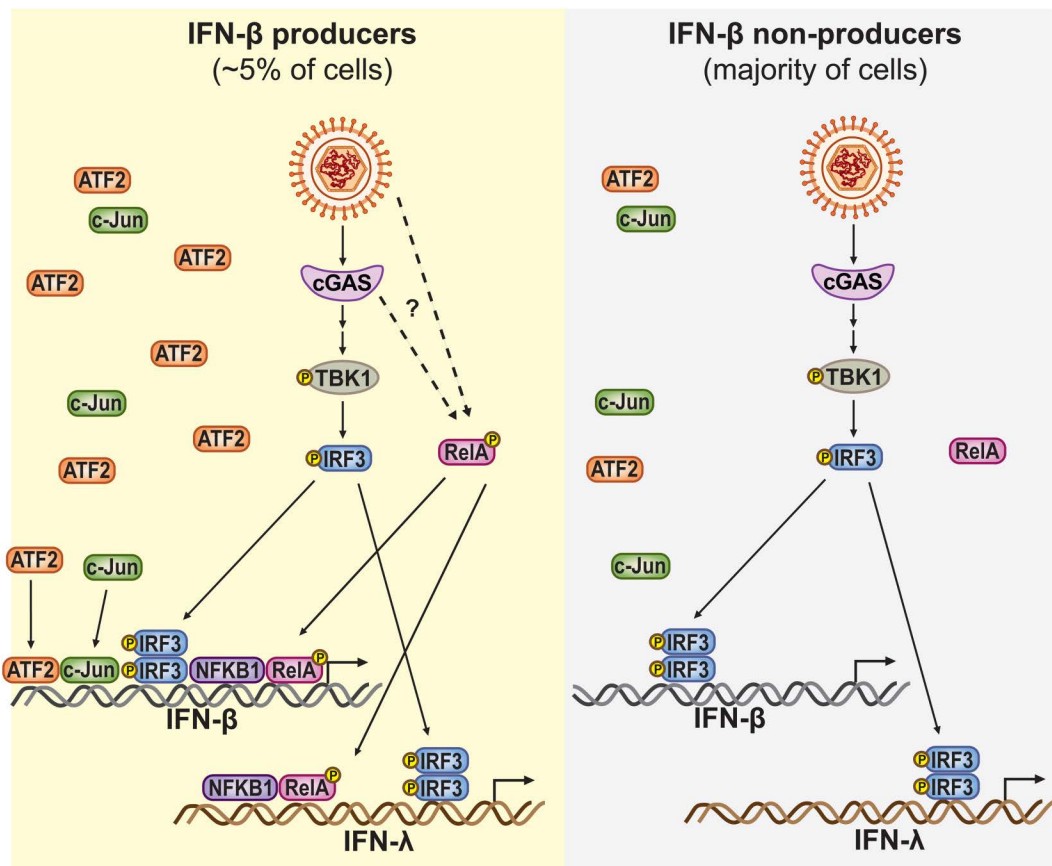

**Fig 6. Model of regulation of heterogeneous IFN-β expression by components of the IFN-β enhanceosome.** (Left) In the rare cells that induce IFN-β during viral infection as "first responders", there are high levels of the AP-1 factor ATF2 and both the NF-κB subunit RelA and IRF3 become phosphorylated in response to infection. This results in the formation of the IFN-β enhanceosome and IFN-β and IFN-λ1 transcription. (Right) In the majority of virally infected cells, there are lower amounts of ATF2, and RelA is not phosphorylated during infection. While IRF3 is still activated, the full IFN-β enhanceosome does not form and IFN-β and IFN-λ1 are not induced.

IRF3, are likely to determine whether individual cells induce IFN-β during KSHV and caspase inhibition. Protein levels of the AP-1 factor ATF2 and phosphorylation of the NF-κB factor RelA were elevated in cells that expressed higher levels of IFN-β compared to cells that expressed low IFN-β. ATF2 levels were not upregulated under IFN-inducing conditions in bulk samples, suggesting that sorting for cells with high IFN-β enriched for cells with high ATF2. Therefore, this result points to a model whereby ATF2 heterogeneity precedes viral reactivation, and only cells with higher ATF2 respond to pathogen cues by turning on IFN-β. In contrast, RelA was only activated under IFN-inducing conditions, which suggests a possible bottleneck to its activation in most cells. Our findings thus support a model in which IFN-β inducibility during KSHV lytic replication is limited to a rare fraction of cells with elevated baseline ATF2 levels and activated NF-κB. While IRF3, AP-1, and NF-κB have long been known to regulate IFN transcription, the cell-to-cell heterogeneity in their levels and activation has not previously been reported. Moreover, this is the first description of a possible role for AP-1 and NF-κB in controlling which individual cells make IFN, despite cGAS-to-IRF3 signaling being active in almost all infected cells in the population.

Host regulation of IFN-β is important given its potent effects on both the host and pathogen through autocrine and para-crine induction of hundreds of ISGs. Insufficient induction of ISGs can lead to susceptibility to infection, while excessive IFN-related inflammation can lead to autoimmune disorders and excessive tissue damage. A system in which the barrier for IFN-β induction is very high may provide an appropriate level of innate immune alarm by limiting IFN-β expression to a small fraction of infected cells. The fact that many viral infections display heterogeneous IFN induction in a way that is not fully explained by viral factors [11,19–21], in combination with our results, suggests that an important part of the mechanism of IFN regulation is rooted in cellular heterogeneity. Furthermore, heterogeneous expression of other cytokines such as IL-2 [50], IL-4 [51,52], IL-5 [53], and IL-10 [54] has been reported, suggesting this is an important mode of immune regulation.

Interestingly, while IRF3 and TBK1 are required for IFN-β induction following PAMP/DAMP signaling, here we found they are not sufficient. Moreover, this result shows that heterogeneity in the presence of the PAMP/DAMP stimulus, which in our system is likely mitochondrial DNA (S3 Fig), cannot explain IFN induction heterogeneity. Although IRF3 activation is often used as a proxy for IFN-β induction, especially in studies of viral infections, our results indicate that this one-to-one relationship is not true at the level of individual cells. Indeed, both TBK1 and IRF3 appear to be activated regardless of the IFN-expressing status of the cells. Most likely, the upstream components of the IFN induction pathway, including the PRR (cGAS in this case), are also activated regardless of the IFN status of the cells. Instead, our results point to the other IFN-β enhanceosome factors, AP-1 and NF-κB, as the potential limiting factors that determine IFN-β induction in individual cells. We propose a model where activation of IRF3 is the main signal that reports on the presence of an infection, while the main function of AP-1 and NF-κB is to provide an additional barrier to limit IFN-β induction to a small subset of cells among an infected population.

The regulation of AP-1 and NF-κB during IFN responses is not as well characterized as the activation of IRF3. For AP-1, our results suggest that differences may precede infection, because at the bulk level there is no clear change in ATF2 levels. Therefore, the simplest interpretation for our results (Fig 5D) is that some cells have higher ATF2, and these cells are more abundant in the IFN-producing pool. We were unable to detect changes in phosphorylation of ATF2 and c-Jun under IFN-inducing conditions (S5A-C Fig), suggesting this activation step may be dispensable in our system. In contrast, NF-κB is likely activated in a subset of cells under IFN-inducing conditions. IRF3 and NF-κB may both be acti-vated downstream of STING based on previous studies [55,56]. Thus, the fact that NF-κB is only phosphorylated in IFN-β high cells, while IRF3 is also phosphorylated in IFN-β low cells, indicates that there must be a heterogenous bottleneck specifically for NF-κB activation. We attempted to use inhibitors of the IκB kinases α and β (IKKα/β), which activate RelA under other conditions [57–59], to test whether they are required for RelA phosphorylation and IFN-β induction in KSHV infection. However, IKKα/β inhibitors strongly inhibited KSHV reactivation, which precluded testing of effects on IFN-β induction. Therefore, we still do not know whether IKKα/β or other kinases phosphorylate NF-κB during IFN-β induction in our system. It is also possible that IRF3 is phosphorylated but does not bind its targets in the nucleus. However, in our

current system, investigating IFN-β promoter occupancy by IRF3, AP-1, and/or NF-κB using ChIP or similar techniques is not feasible because the sorting does not yield a sufficient number of tdTomato-positive cells even after multiple hours.

While this study has shed light on the expression mechanics of IFN-β during KSHV infection, it is possible that the mechanism of IFN heterogeneity is different for other viral infections and cell types. Although IFN heterogeneity is a widespread feature of viral infection, reported by every single-cell study we have encountered, the PRRs and signaling pathways that lead to IFN induction vary [6–18,60]. Moreover, IFN-β expression patterns appear to differ between cell types. For example, a larger proportion of dendritic cells express IFN-β upon infection [19,61,62] compared to other cell types such as fibroblasts and epithelial cells [6–16]. Flexibility in tailoring the IFN response to factors such as cell type, viral load, and PAMP type is likely evolutionarily beneficial in mounting an appropriate immune response. To note, a recent study using poly(I:C) to investigate heterogeneity corroborated our finding that TBK1 and IRF3 are activated regardless of IFN-producing status of the cells [63]. More studies identifying the determinants of IFN expression are needed to parse out the ubiquity vs. uniqueness of our findings.

Although our findings suggest there is a high barrier to IFN-β induction, results from studies using Sendai virus suggest that heterogeneity of IFN-β induction is not due to a complete inability of some cells to induce IFN-β. Infection with a highly inflammatory strain of Sendai virus (Cantell) induced IFN-β expression in 99% of infected A549 cells at high multiplicity of infection [11,62]. In comparison, only 0.48% of A549 cells expressed IFN-β during infection with influenza A virus [11]. This indicates that IFN-β regulation can be overridden given excessive stimuli such as the defective-interfering viral genomes that characterize the Sendai virus strain Cantell [64]. These observations also suggest that the strength or abundance of the stimulus may matter. Indeed, strong IFN inducers like Newcastle disease virus, Sendai virus strain Cantell, or the potent PAMP mimetic poly(I:C) induce IFN-β in a larger proportion of cells (10–99%) [20,21,60,61,63,65–68] compared to other viral infections and treatments (1–10%) [6–17]. Unfortunately, we have been unable to test whether the relationship between stimulus strength and percentage of IFN-expressing cells is also true in KSHV-reactivating BC-3 cells, as we have not identified a stronger stimulus for these cells. To isolate the cells that express IFN-β, the cells need to be active and viable for long enough for the tdTomato fluorescent protein to express and accumulate to detectable levels. Commonly used IFN-inducing stimuli (poly(I:C), 2'3'-cGAMP, STING agonist diABZI, TLR7 agonist imiquimod, and plasmid DNA) either did not induce IFN-β expression in BC-3 cells or induced cell death before we could measure tdTomato production. IFN-β activation through KSHV reactivation and caspase inhibition allows for sustained IFN-β induction without triggering cell death, which gives us a unique tool to study the elusive phenomena of IFN heterogeneity.

While our scRNA-seq data did not reveal any correlations between viral genes and IFN-β expression [6], technical limitations of short read sequencing may have missed some important components. For example, the scRNA-seq would not have captured mutations in KSHV that may have arisen in individual cells. Moreover, the sequencing method was 3' end directed, and many mRNAs in KSHV share 3' end sequences [69], preventing accurate measurement of some viral genes. KSHV has evolved a large arsenal of proteins that antagonize IFN signaling. KSHV ORF52 can prevent cGAS activation by sequestering DNA substrates by phase separation mechanism [70], and a truncated and therefore cytoplasmic form of KSHV LANA also inhibits cGAS [71]. KSHV ORF33 aids in inactivating the cGAS-adaptor STING [72]. One study found that the KSHV homolog of human IRFs vIRF1 inhibits IFN-β by preventing STING and TBK1 activation during KSHV reactivation in iSLK.219 cells [73]. However, these viral factors all act upstream of IRF3 activation, which still occurs even in IFN-β-low cells in our system, and are therefore unlikely to explain IFN induction heterogeneity. KSHV vIRFs have also been implicated in regulating IFNs at the transcriptional level. Studies on KSHV vIRF3 and vIRF4 have found that these inhibit IRF7, which is important for inducing secondary responder cells [74,75]. A study on KSHV vIRF2 found that it induces IRF3 degradation via caspase-3 [76]. Another study found KSHV vIRF1 inhibits IFNA4 induced by Newcastle disease virus in murine cells by preventing p300 from binding to IRF1 and IRF3 [77]. KSHV vIRF1 also binds CBP/p300 to prevent IRF3 complex formation on the IFN-β promoter during Sendai virus infection in 293T cells [78]. Potentially, these events could create a barrier downstream of IRF3 phosphorylation, but this remains to be tested. Also, there has been no

PLOS Pathogens

reports of any KSHV factor that regulates AP-1 levels or blocks NF-κB activation, so there is no obvious viral explanation for the heterogeneity we observe for these factors.

There are some limitations to our study. First, our reporter system likely does not fully recapitulate native IFN-β mRNA expression and kinetics. The minimal IFN-β promoter lacks the native chromatin environment and potential long range promoter elements. The reporter also lacks the native 3' untranslated region of the IFN-β mRNA, which makes this mRNA unstable. Also, it is likely that only cells with sustained IFN-β promoter activity were detected and sorted, as sufficient accumulation of the tdTomato protein is needed for FACS. However, despite these imperfections, we were able to successfully separate cells that expressed high and low levels of IFN-β at the time of sorting (Fig 1D). Also, while our results point to cellular heterogeneity as a major determinant of IFN-β regulation, we cannot completely rule out the possibility that the heterogeneity of AP-1 and NF-κB is regulated by the virus, as described above. Lastly, while TPA is a very effective inducer of lytic KSHV reactivation and is commonly used in KSHV research [35], it is also a known inducer of NF-κB. This is a potentially confounding variable. Our concerns are reduced by the fact that treatments that induce IFN-β in infected BC-3 cells do not induce IFN-β, IFN-λ1, or phosphorylation of TBK1, IRF3, or RelA in the uninfected BJAB cells at comparable time points, indicating viral reactivation rather than TPA treatment drives signaling changes in our system (Figs 1H, 1I, 4M, 5G, and 5J). Moreover, to date, all our results are consistent between the BC-3 and iSLK.219 KSHV-infected cell lines [6], even though iSLK.219 cells are not treated with TPA to induce the lytic cycle. This also suggests that our results are not simply due to TPA effects.

Despite the described limitations, this study points to a new model of heterogeneous type I IFN transcription during viral infection as the result of inherent cellular heterogeneity in protein levels and activation. More studies are needed to pinpoint the earliest source of the heterogeneity and to examine the mechanism of enrichment and activation of AP-1 and NF-κB in the context of antiviral responses.

## Materials and methods

### Plasmids

pLJM1_AP-1mut_2xIFNBp-tdTomato, pLJM1_IRF3mut_2xIFNBp-tdTomato, and pLJM1_NF-κBmut_2xIFNBp-tdTomato constructs were generated from pLJM1_2xIFNBp-tdTomato [6] by introducing mutations in the AP1, IRF3, and NF-κB binding sites. The mutations were designed based on previously published studies that showed loss of IFN-β induction [79–81]. pLJM1-tdTomato construct was generated from pLJM1-GFP, a gift from David Sabatini (Addgene plasmid number 19319; http:// n2t.net/addgene:19319; RRID, Addgene_19319) [82]. All constructs were generated by Gibson cloning with HiFi assembly reagent (New England Biolabs, E2621)

### Cell lines, reagents, and treatments

All cells were cultured at 37°C and 5% $CO_2$. A549 and HEK293T cells were grown in Dulbecco's modified Eagle's medium (DMEM; Life Technologies) supplemented with 10% fetal bovine serum (FBS; Hyclone). BJAB, BC-3, and BC-3 derivative cells were maintained at a density of $5 \times 10^5$ cells/mL in Roswell Park Memorial Institute medium (RPMI; Life Technologies) supplemented with 20% FBS, 2 mM GlutaMAX (Gibco), and 55 μM β-mercaptoethanol (BME; Gibco). BC-3-IFN-βp-tdTomato cells were generated as described in Lent, Tabtieng, et al., 2022 [6]. BC-3-CMVp-tdTomato, BC-3-IFN-βp-tdTomato AP-1mut, IRF3mut, and NF-κBmut were generated by transducing BC-3 cells with pLJM1-tdTomato, pLJM1_AP-1mut_2xIFNBp-tdTomato, pLJM1_IRF3mut_2xIFNBp-tdTomato, and pLJM1_NFκBmut_2xIFNBp-tdTomato. Lentiviral packaging was carried out using packaging plasmids pMDLg/pRRE (Addgene plasmid number 12251; http://n2t.net/addgene:12251; RRID, Addgene_12251), pRSV-Rev (Addgene plasmid number 12253; http://n2t.net/addgene:12253; RRID, Addgene_12253), and pMD2.G (Addgene plasmid number 12259; http://n2t.net/addgene:12259; RRID, Addgene_12259), kind gifts from Didier Trono [83]. The transduced cells were not clonally selected, because

KSHV reactivation can be variable among KSHV-infected cells in a population and clonal selection can result in artifacts. For reactivation or reactivation-like treatment, BJAB, BC-3 and BC-3-derivative cells were seeded at a density of $5\times10^5$ cells/mL and treated with 20 ng/mL TPA (12-O-tetradecanoylphorbol-13-acetate; MilliporeSigma 5244001MG) in dimethylsulfoxide (DMSO; Sigma Aldrich Fine Chemicals Biosciences). Where indicated, cells were treated with vehicle (DMSO), 10 µM IDN-6556 (Emricasan; Selleck Chemical LLC S777525MG), a mixture of neutralizing antibodies against type I IFNs at 1:2000 dilution (PBL Assay Science 39000–1), 10 µM TBK1 inhibitor (MRT67307; Medchemexpress LLC HY130185MG), 10 µM cGAS inhibitor (G140; Invivogen inh-g140), and/or 1 µM dual AP-1 and NF-κB inhibitor (SP100030, MilliporeSigma Calbiochem, 5315350001). To deplete BC-3 cells of mitochondrial DNA, cells were incubated in 20 µM 2'3'-dideoxycytidine (ddC, Thermo Fisher, L10619.MF) for 48 hours before viral reactivation experiments with TPA. Where indicated, A549 cells were seeded at $0.5\times10^6$ cells per well in a 6-well plate 1 day before treatment, and treated with vehicle (water) or 7 µg/mL poly(I:C) (poly(I:C) LMW/ LyoVec, Invivogen, TLRLPICWLV) for 6 or 48 hours.

## Fluorescence-activated cell sorting

100 mL of BC-3-IFN-βp-tdTomato or BC-3-CMVp-tdTomato cells were seeded at a density of $5\times10^5$ cells/mL and treated with TPA, IDN-1665, and, where indicated, an anti-IFN antibody mixture as described above. After two days, they were collected and stained with 1 µg/mL DAPI (4',6-Diamidino-2-Phenylindole, Dihydrochloride) stain (Invitrogen, D1306) for 5 minutes at room temperature. They were then washed in phosphate buffer saline (PBS; Life Technologies), filtered through a 35 µm strainer (Chemglass Life Sciences CLS4380009) and sorted based on tdTomato fluorescence on a BD FACSAria III at the University of Wisconsin Carbone Cancer Center Flow Cytometry Laboratory core facility. The DAPI stain was used to remove dead cells from the analysis, as in the absence of permeabilization it only stains dead cells. Gates for tdTomato fluorescence were drawn based on a negative control consisting of BC-3 cells that lack the reporter. BC-3 cells lacking the reporter were treated in the same way as the sorted reporter cells. In each IFN-β reporter experiment using BC-3-IFN-βp-tdTomato cells, about $80\times10^6$ cells were sorted, and $3\times10^6$ tdTomato+ and $12\times10^6$ tdTomato- cells were collected. In each control experiment using BC-3-CMVp-tdTomato cells, about $10\times10^6$ cells were sorted, and $2\times10^6$ tdTomato+ and $2\times10^6$ tdTomato- cells were collected. Sorted cells were pelleted and resuspended in 1 mL of PBS. 100 µL of cell suspension were used for RNA extraction and 900 µL for protein extraction. Bulk controls were prepared in parallel using 4 mL of BC-3-IFN-βp-tdTomato or BC-3-CMVp-tdTomato cells seeded at a density of $5\times10^5$ cells/mL and treated with TPA, IDN-6556, and/or anti-IFN antibodies (or corresponding vehicles), as indicated for 2 days. The bulk control cells were pelleted and resuspended in 1 mL of PBS. 50 µL of cell suspension were used for RNA extraction, 450 µL for protein extraction, and 500 µL for flow cytometry to determine KSHV reactivation efficiency.

## RT-qPCR and qPCR

RNA and DNA samples were collected from BJAB cells, BC-3 cells, or BC-3 derivatives treated as described in the Cell lines, reagents, and treatments section. For experiments involving sorting, a fraction of the sorted cells or bulk samples was used as described in the Fluorescence-activated cell sorting section. For experiments not involving sorting, 4 mL of cells were collected as described in the same section for bulk control samples. In both cases, for RNA only extraction cells were pelleted by centrifugation and resuspended in RNA lysis buffer (Zymo research). Total RNA was extracted using the Quick-RNA MiniPrep kit (Zymo research) following the manufacturer's protocol. For DNA and RNA collection (mitochondrial DNA depletion experiments only), cells were pelleted by centrifugation and resuspended in DNA/RNA lysis buffer (Zymo research). Total DNA and RNA were extracted using the Quick-DNA/RNA MiniPrep Plus kit (Zymo research) following the manufacturer's protocol. For human mRNAs, cDNA was prepared using an iScript cDNA synthesis kit (Bio-Rad) following the manufacturer's protocol. For KSHV genes, cDNA was prepared using AMV RT (Promega) following the manufacturer's protocol using 1 pmol of human 18S reverse primer and 1 pmol reverse primer for each target to select for transcripts from the correct strand of the KSHV genome. For mitochondrial and nuclear DNA measurements, purified DNA was used as

template for qPCR. qPCR was performed using iTaq Universal SYBR green Supermix (Bio-Rad) following the manufacturer's protocol on the CFX Duet Real-Time PCR System (Bio-Rad). No-template and, where applicable, no-RT controls were included in each replicate experiment. Experiments were only included in the analysis if these controls showed negligible amplification rates. In all cases, target mRNA levels were normalized to levels of 18S rRNA as an internal standard. Mitochondrial DNA levels were normalized to nuclear CCR5 DNA. CFX Maestro software was used to analyze the data.

Primers used are listed in S1 Table [84–99].

## Protein analysis

Protein samples were collected from BC-3-IFN-βp-tdTomato, BJAB, and A549 cells treated as described in the Cell lines, reagents, and treatments section. For experiments involving sorting, a fraction of the sorted cells was used as described in the Fluorescence-activated cell sorting section. For samples not involving in sorting, 4 mL of BJAB or BC-3-IFN-βp-tdTomato cells were used and collected as described for bulk control samples in that section. In both cases, cells to be lysed were pelleted by centrifugation and resuspended in an NP-40-only buffer (50 mM Tris-HCl pH 7.4, 150 mM NaCl, 1 mM EDTA, 0.5% NP-40) supplemented with a broad-spectrum protease inhibitor mixture (Thermo Scientific A32955) and a broad-spectrum phosphatase inhibitor mixture (Thermo Scientific A32957). Protein concentration was determined by Bradford method (Bio-Rad 5000006). Between 10 and 20 μg of protein were loaded in each well. Laemmli buffer (Bio-Rad) was added to samples before incubation at 95°C for 5 min. Samples were separated by SDS-PAGE and transferred to polyvinylidene difluoride membranes (PVDF; Fisher Scientific IPFL00010) using the semi-dry Trans-Blot Turbo Transfer System (Bio-Rad). Membranes were blocked in 5% bovine serum albumin (BSA; Fisher BioReagents BP1600–100) in Tris-Buffer Saline with 0.1% Tween (TBST). All antibodies used were diluted 1:1000 in 5% BSA in TBST for staining. The following Cell Signaling Technologies antibodies were used: IRF3 (D6I4C, no.11904), phospho-IRF3 Ser386 (E7J8G, no. 37829), phospho-IRF3 Ser396 (4D4G, no. 4947), TBK1 (D1B4, no. 3504), phospho-TBK1 Ser172 (D52C2, no. 5483), ATF2 (D4L2X, no.35031), phospho-ATF2 (E3D6B, no. 44167), c-Jun (60A8, no. 9165), phospho-c-Jun (Ser63, no. 9261), NF-κB p65/RelA (D14E12, no. 8242), phospho-NF-κB p65/RelA Ser536 (93H1, no. 3303). In addition, antibodies against β-actin (Abcam, ab8229) were used to stain for loading controls. Secondary horseradish peroxidase (HRP)-conjugated antibodies (goat anti-rabbit and rabbit anti-goat IgG) were purchased from Southern Biotechnology (no.4030-05, no.OB6106-05) and used at 1:5000 in 5% BSA in TBST. All membranes were exposed using SuperSignal West Pico PLUS Chemiluminescent Substrate (Thermo Scientific) and imaged with an iBright FL1000 imaging system (Thermo Scientific). Protein quantification from western images was done using Image J [100].

## Flow cytometry analysis

BC-3, BC-3-IFN-βp-tdTomato, or BC-3-CMVp-tdTomato cells were fixed with 4% paraformaldehyde (PFA; Electron Microscopy Sciences) in PBS for 15 min at room temperature and washed twice with PBS. Then, the cells were permeabilized by incubating in ice cold 90% methanol diluted in 0.5% BSA in PBS for 5 min on ice and washed twice with 0.5% BSA in PBS. For assessment of reactivation efficiency, cells were stained for a KSHV lytic infection marker, ORF45. They were incubated with antibodies against KSHV ORF45 (Thermo Fisher MA514769) at 1:100 dilution for 1 hr at room temperature, washed twice, and then stained with an Alexa Fluor 647 (AF647)-conjugated secondary antibody at 1:500 dilution (Thermo Fisher A21236) for 30 min at room temperature in the dark. 0.5% BSA in PBS was used for all incubations and washes. Cells were collected by centrifugation at 500 x $g$ for 5 min between all washes/incubation. After two washes, cells were resuspended in 0.5% BSA in PBS and filtered through a 35 μm strainer (Chemglass Life Sciences CLS4380009) for analysis. The percentage of tdTomato+ or ORF45+cells was quantified by flow cytometry on a Thermo Fisher Attune NxT V6 cytometer at the UW-Madison Carbone Cancer Center Flow Cytometry Laboratory. AF647 fluorescence was gated based on the corresponding latently infected and vehicle (DMSO)-treated cells. $3 \times 10^6$ events were collected per sample. FlowJo 10.8.2 was used for the data analysis.

### Single cell RNA sequencing analysis

A dataset obtained from iSLK.219 cell samples previously published in Lent, Tabtieng, et al., 2022 [6] (GSE190558) was re-analyzed in the current study by extracting prepared Seurat matrices and manually selecting genes of interest.

### Statistics

All statistical analysis was performed using GraphPad Prism version 10.3.0 or later (GraphPad Software, Boston, MA USA; www.graphpad.com). Statistical significance was determined by one- or two-way ANOVA followed by a *post hoc* multiple-comparison test (Dunnett or Tukey) when multiple comparisons were required. Statistical analysis of qRT-PCR data was performed on $\log_2$ transformed data to ensure normal distribution [101]. Singular outliers were identified using Grubbs' test (alpha = 0.05) for outliers, and removed from Figs 1G, 2A, 3D, 3G, S1A Fig, S1E Fig, and S4D Fig. All plots represent mean +/- standard deviation. Figure legends indicate the number of biological replicates used for each experiment. All western blot images are representative of 3 or more independent biological replicates.

### Supporting information

**S1 Fig. Viral gene expression heterogeneity does not completely explain IFN-β expression.** BC3-IFN-βp-tdTomato reporter cells were treated with 20 ng/ml TPA for 48 hours to induce the lytic cycle ("lytic (+TPA)"), 10 μM of the pan-caspase inhibitor IDN-6556 ("casp-i"), and/or a cocktail mix of antibodies against type I IFNs and their receptor ("anti-IFN Abs") at 1:2000 dilution. Cells from the lytic+casp-i (**A, C, E, G, I, K**) or the lytic+casp-i+anti-IFN Abs (**B, D, F, H, J, L**) samples were collected both without sorting ("bulk") and after sorting based on tdTomato expression (tdTomato + and -). RNA was extracted and KSHV mRNA levels were measured by RT-qPCR and normalized to 18S rRNA. Immediate early genes: ORF50 (**A-B**), ORFK5 (**C-D**). Early genes: ORF57 (**E-F**), ORF59 (**G-H**). Late genes: ORF42 (**I-J**), and ORF52 (**K-L**). The fold enrichment of mRNA levels compared to unsorted ("bulk") samples are plotted in $\log_2$ scale. Results in this figure were obtained from the same samples used in Figs 1 and 2 and are thus directly comparable to IFN-β enrichment shown in those Figure. ns = p > 0.05, ** = p < 0.01. One-way ANOVA followed by Tukey's post hoc multiple comparisons test. n ≥ 4 for all experiments. (EPS)

**S2 Fig. CMV promoter-driven tdTomato expression does not induce IFN-β in BC-3 cells and does not correlate with IFN-β expression. A.** Diagram of CMV promoter-driven tdTomato sorting strategy. A cassette containing the tdTomato fluorophore driven by a CMV promoter was introduced in BC-3 cells via lentiviral transduction for stable integration. RNA was collected from cells sorted based on tdTomato expression. **B-E.** BC3-CMVp-tdTomato cells were treated with 20 ng/mL TPA for 48 hours to induce the lytic cycle ("lytic (+TPA)"). Where indicated, the cells were also treated with 10 μM of the pan-caspase inhibitor IDN-6556 ("casp-i"). **B.** Cells were analyzed by flow cytometry for tdTomato fluorescence. **C.** Cells were stained for KSHV ORF45 and analyzed by flow cytometry. **D-E.** RNA was isolated from cells collected without sorting ("bulk") or after sorting based on tdTomato expression (tdTomato+ and -). IFN-β and tdTomato mRNA levels were measured by RT-qPCR and normalized to 18S rRNA. mRNA levels are plotted relative to the lytic+casp-i bulk sample. ns = p > 0.05, ** = p < 0.01, *** = p < 0.001, **** = p < 0.0001. One-way ANOVA followed by Tukey's (B) or Dunnett's (C) post hoc multiple comparisons test. Two-way ANOVA followed by Tukey's post hoc multiple comparisons test (D, E). n = 3 for all experiments. (EPS)

**S3 Fig. Mitochondrial DNA is the likely cGAS ligand during lytic KSHV infection.** BC-3 cells were treated with or without 20 μM dideoxycytidine ("ddC") for 48 hours to deplete mitochondrial DNA ("mtDNA"), then treated with 20 ng/ml TPA for 48 hours to induce the lytic cycle ("lytic (+TPA)"), or vehicle (DMSO, "latent"). At the time of lytic induction, where indicated, the cells were also treated with 10 μM of the pan-caspase inhibitor IDN-6556 ("casp-i"). **A.** DNA was extracted

and mtDNA levels were measured by qPCR, normalized to CCR5 nuclear DNA, and plotted relative to the untreated lytic+casp-i sample. **B.** RNA was extracted and IFN-β mRNA levels were measured by RT-qPCR, normalized to 18S rRNA, and plotted relative to the untreated lytic+casp-i sample. **C.** Cells were stained for KSHV ORF45 and analyzed by flow cytometry. $ns = p > 0.05$, $** = p < 0.01$, $**** = p < 0.0001$. Two-way ANOVA followed by Tukey's post hoc multiple comparisons test for all experiments. $n = 3$ for all experiments.
(EPS)

**S4 Fig. IFN-β induction during KSHV lytic reactivation requires all three IFN-β enhanceosome transcription factors. A-B.** BC-3 cells were treated with 20 ng/ml TPA for 48 hours to induce the lytic cycle ("lytic (+TPA)"), 10 μM of the pan-caspase inhibitor IDN-6556 ("casp-i"), and where indicated, 1 μM of the dual AP-1 and NF-κB inhibitor SP100030. **A.** RNA was extracted and IFN-β mRNA levels were measured by RT-qPCR, normalized to 18S rRNA, and plotted relative to the sample treated with TPA, casp-i, and vehicle control ("(-)"). **B.** Cells were stained for KSHV ORF45 and analyzed by flow cytometry. **C.** Diagram of the IFN-β promoter used in the reporter constructs, including mutations inserted to generate enhanceosome component binding mutants. Cassettes containing these mutant promoters were introduced in BC-3 cells via lentiviral transduction for stable integration. **D-E.** BC-3 cells with stably integrated tdTomato reporters driven by the promoters shown in **C** were treated with 20 ng/ml TPA for 48 hours to induce the lytic cycle ("lytic (+TPA)") and 10 μM of the pan-caspase inhibitor IDN-6556 ("casp-i"). **D.** RNA was extracted and tdTomato mRNA levels were measured by RT-qPCR and normalized to 18S rRNA. mRNA levels are plotted relative to the cells containing the WT promoter-driven reporter, lytic+casp-i sample. **E.** Cells were stained for KSHV ORF45 and analyzed by flow cytometry. $ns = p > 0.05$, $*** = p < 0.001$, $**** = p < 0.0001$. Two-way ANOVA followed by Dunnett's post hoc multiple comparisons test for all experiments. $n = 5$ for A-B, $n = 4$ for D-E.
(EPS)

**S5 Fig. AP-1 and NF-κB may be more active in IFN-β high cells.** BC3-IFN-βp-tdTomato reporter cells were treated with 20 ng/ml TPA for 48 hours to induce the lytic cycle ("lytic (+TPA)"), 10 μM of the pan-caspase inhibitor IDN-6556 ("casp-i"), and a cocktail mix of antibodies against type I IFNs and their receptor ("anti-IFN Abs") to block IFN-β paracrine signaling. **A-C.** Protein was isolated from BC3-IFN-βp-tdTomato reporter cells without sorting ("bulk"), or after sorting the lytic+casp-i+ anti-IFN Abs based on tdTomato expression. Western blots were probed for phosphorylated ATF2, phosphorylated c-Jun, and β-actin as a loading control. Levels of phosphorylated ATF2 (**B**) and phosphorylated c-Jun (**C**) in sorted samples were quantified from the western blots, normalized to β-actin, and plotted relative to bulk lytic+casp-i+ anti-IFN Abs samples. **D-E.** RNA was extracted from BC3-IFN-βp-tdTomato reporter cells without sorting ("bulk"), or after sorting the lytic+casp-i+anti-IFN Abs based on tdTomato expression (tdTomato+ and -). mRNA levels of targets of AP-1 – ATF3, GADD45B, and HSPA1A (**D**) – and targets of AP-1 and NF-κB – IL-6 and TNFAIP3 (**E**) – were measured by RT-qPCR, normalized to 18S rRNA, and plotted relative to the bulk lytic+casp-i+anti-IFN Abs sample. $ns = p > 0.05$, $* = p < 0.05$, $** = p < 0.01$. One-way ANOVA followed by Tukey's post hoc multiple comparisons test for all experiments. $n = 3$ for A-C, $n = 4$ for D-E.
(EPS)

**S1 Table. Primers used for qPCR in this study.**
(DOCX)

# Acknowledgments

We thank members of the Gaglia laboratory for suggestions and feedback on the project and the manuscript. We thank the University of Wisconsin Carbone Cancer Center (UWCCC) Flow Cytometry Laboratory staff for technical and conceptual assistance and services. The UWCCC Flow Cytometry Laboratory is supported by the National Institutes of Health UWCCC Cancer Center Grant P30 CA014520. We thank D. Trono and D. Sabatini for sharing reagents.

## Author contributions

**Conceptualization:** Machika Kaku, Marta Maria Gaglia.

**Data curation:** Machika Kaku, Marta Maria Gaglia.

**Formal analysis:** Machika Kaku.

**Funding acquisition:** Marta Maria Gaglia.

**Methodology:** Machika Kaku.

**Project administration:** Machika Kaku, Marta Maria Gaglia.

**Resources:** Machika Kaku, Marta Maria Gaglia.

**Software:** Machika Kaku.

**Supervision:** Marta Maria Gaglia.

**Validation:** Machika Kaku.

**Visualization:** Machika Kaku.

**Writing – original draft:** Machika Kaku, Marta Maria Gaglia.

**Writing – review & editing:** Machika Kaku, Marta Maria Gaglia.

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
