## [Editor Report · Decision Letter 0]

18 Nov 2025

Interferon-β induction heterogeneity during KSHV infection is correlated to levels and activation of the transcription factors ATF2 and RelA, and not IRF3

PLOS Pathogens

Dear Dr. Gaglia,

Thank you for submitting your manuscript to PLOS Pathogens. After careful consideration, we feel that it has merit but does not fully meet PLOS Pathogens's publication criteria as it currently stands. Therefore, we invite you to submit a revised version of the manuscript that addresses all the points raised during the review process at Review Commons.

We look forward to receiving your revised manuscript.

Kind regards,

Pinghui Feng

Academic Editor

PLOS Pathogens

Blossom Damania

Section Editor

PLOS Pathogens

Editor-in-Chief

PLOS Pathogens

orcid.org/0000-0003-2946-9497

Editor-in-Chief

PLOS Pathogens

orcid.org/0000-0002-7699-2064

**Journal Requirements:**

https://journals.plos.org/plospathogens/s/submission-guidelines#loc-parts-of-a-submission

3) We noticed that you used the phrases 'data not shown' and 'unpublished data' in the manuscript. We do not allow these references, as the PLOS data access policy requires that all data be either published with the manuscript or made available in a publicly accessible database. Please amend the supplementary material to include the referenced data or remove the references.

**Reviewers' Comments:**

**Figure resubmission:**

**Reproducibility:**



---

## [Decision Letter · Decision Letter 1]

28 Jan 2026

Dear Dr. Gaglia,

We are pleased to inform you that your manuscript 'Interferon-β induction heterogeneity during KSHV infection is correlated to levels and activation of the transcription factors ATF2 and RelA, and not IRF3' has been provisionally accepted for publication in PLOS Pathogens.

As a personal note, I appreciate the significance of your findings described here as the two reviewers. I would like to point out that we previously reported that deamidation can dictate the gene-specific activation of RelA and IRF3, although conventional assays measuring RelA and IRF3 activation (e.g., phosphorylation and nuclear translocation) will not show apparent difference (Cell Metabolism, 2020; Immunity, 2025). These may provide some alternatives to explain the phenotypes your paper describes here.

Best regards,

Pinghui Feng

Academic Editor

PLOS Pathogens

Blossom Damania

Section Editor

PLOS Pathogens

Sumita Bhaduri-McIntosh

Editor-in-Chief

PLOS Pathogens

orcid.org/0000-0003-2946-9497

Michael Malim

Editor-in-Chief

PLOS Pathogens

orcid.org/0000-0002-7699-2064

Reviewer Comments (if any, and for reference):

Reviewer's Responses to Questions

**Part I - Summary**

Reviewer #1: The authors have been very responsive to reviewer concerns and nicely strengthened their main conclusions.

Reviewer #2: Main Finding: This study shows that the difference in IFN-β production during KSHV infection is determined by the levels and activation of host factors ATF2 and RelA, rather than IRF3. Using a reporter system, the authors demonstrate that even though IRF3 is activated in most infected cells, it is not the limiting factor. Instead, only a small subset of cells that have higher ATF2 protein levels and RelA phosphorylation are able to produce IFN-β.

Significance: This work explains the mechanism of why only a small percentage of cells produce type I interferon during viral infection. This uneven response is observed in many viral infections, but the reason was not well understood before. This study identifies ATF2 and RelA as the key factors underpinning this difference, which advances our understanding on how the innate immune system works at the single-cell level.

Comments:

1) Response to Reviewers: The authors answered all questions very well.

To address the concern about PAMP heterogeneity, they added Figure S3. This shows mitochondrial DNA is likely the ligand and proves that PAMP variation is not the main cause of the difference.

For the IRF3 activation markers, the authors added new data for IRF3 Ser396 phosphorylation (Fig 4F-G). This confirms that IRF3 activation is similar in both IFN-high and IFN-low cells, supporting the claim that IRF3 activation is not the bottleneck.

Also, the authors created a CMV promoter control line (Fig S2). This effectively rules out artifacts from random lentiviral integration.

2) Manuscript Structure: The structure of the manuscript is clear and the logic is strong. It starts by validating the reporter system, then systematically excludes stochastic noise and the TBK1-IRF3 pathway. At the end, it identifies ATF2 and RelA as the drivers of IFN induction. The addition of downstream target analysis (like ATF3 and TNFAIP3 in Fig S5D-E) further supports their conclusion.

3) Overall Assessment: The authors put a lot of effort into this revision. The new control experiments and biochemical data directly address the main concerns from the reviewers. The data is solid and the manuscript is now very complete.

**Part II – Major Issues: Key Experiments Required for Acceptance**

Reviewer #1: N/A

Reviewer #2: NA

**Part III – Minor Issues: Editorial and Data Presentation Modifications**

Reviewer #1: N/A

Reviewer #2: NA

PLOS authors have the option to publish the peer review history of their article (what does this mean? ). If published, this will include your full peer review and any attached files.

**Do you want your identity to be public for this peer review?** For information about this choice, including consent withdrawal, please see our Privacy Policy .

Reviewer #1: No

Reviewer #2: **Yes:** XINCHI XIE

---

## [Editor Report · Acceptance letter]

Dear Dr Gaglia,

We are delighted to inform you that your manuscript, "

Interferon-β induction heterogeneity during KSHV infection is correlated to levels and activation of the transcription factors ATF2 and RelA, and not IRF3," has been formally accepted for publication in PLOS Pathogens.

Best regards,

Sumita Bhaduri-McIntosh

Editor-in-Chief

PLOS Pathogens

orcid.org/0000-0003-2946-9497

Michael Malim

Editor-in-Chief

PLOS Pathogens

orcid.org/0000-0002-7699-2064